# The high incidence of severe adverse events due to pyrazinamide in elderly patients with tuberculosis

Byoung Soo Kwon[1], Youlim Kim[1,2], Sang Hoon Lee[1,3], Sung Yoon Lim[1], Yeon Joo Lee[1], Jong Sun Park[1], Young-Jae Cho[1], Ho Il Yoon[1], Choon-Taek Lee[1], Jae Ho Lee[1]*

1 Division of Pulmonary and Critical Care Medicine, Department of Internal Medicine, Seoul National University Bundang Hospital, Seongnam-Si, Gyeonggi-Do, South Korea, 2 Division of Pulmonary and Critical Care Medicine, Department of Internal Medicine Chuncheon Sacred Heart Hospital, Chuncheon-si, Gangwon-do, South Korea, 3 Division of Pulmonology, Department of Internal Medicine, Severance Hospital, Institute of Chest Diseases, Yonsei University College of Medicine, Seoul, South Korea

* jhlee7@snubh.org

## Abstract

### Background

Pyrazinamide (PZA) is a common drug that causes serious adverse events (SAEs). The aim of this study was to determine the incidence of and risk factors for SAEs due to PZA during first-line anti-tuberculosis treatment.

### Methods

The medical records of patients with tuberculosis (TB) treated with PZA-containing regimens including first-line drugs—ethambutol, rifampicin, and isoniazid—from January 2003 to June 2016 were reviewed. SAEs were defined as side effects that led to drug discontinuation. The causative drug was determined based on the disappearance of the SAEs upon drug withdrawal and/or the recurrence of the same SAEs with re-challenge.

### Results

Of 2,478 patients with TB, 16.4% experienced SAEs. The incidence of SAEs increased significantly as age increased, except with rifampin. PZA accounted for most SAEs (55.8%). Hepatotoxicity was the most common SAE due to PZA (44.5%), followed by gastrointestinal (GI) intolerance (23.8%). The risk of SAEs due to PZA increased significantly as age increased, when sex and comorbidities were adjusted (odds ratio, 1.013; 95% confidence interval, 1.004–1.023; $P = 0.007$). In the subgroup analysis, older age was an independent risk factor for GI intolerance but not for hepatotoxicity.

### Conclusion

PZA was the most common drug associated with SAEs among the first-line anti-TB drugs, and old age was an independent factor for SAE occurrence. This study suggests that the

**Data Availability Statement:** All relevant data are within the paper and its Supporting Information files.

**Funding:** The authors received no specific funding for this work.

**Competing interests:** The authors have declared that no competing interests exist.

early recognition of whether the causative agent is PZA may improve effective treatment compliance, particularly in elderly patients.

## Introduction

Tuberculosis (TB) is a major cause of death worldwide [1]. Although a standard oral regimen can successfully treat up to 80% of drug-susceptible strains [2], TB is a significant threat to public health systems because of the difficulties in early detection and the required treatment duration [3, 4]. The current guidelines for TB treatment recommend that standard oral drugs—isoniazid (INH), rifampin (RIF), ethambutol (EMB), and pyrazinamide (PZA)—be administered at least for 6 months, with an initial intensive phase followed by a continuation phase [5]. During the treatment period, approximately 8–85% of patients experience various side effects, which range from mild to severe [6]. Severe and fatal drug reactions, or serious adverse events (SAEs) [7, 8], occur in approximately 10–25% of patients who experience side effects [9, 10]. Patients who take medication irregularly or discontinue treatment owing to adverse effects require a longer duration of therapy and are at risk of treatment failure, relapse, or the development of resistance [8, 11–14].

As societal aging progresses, the development of TB in the elderly is increasing [15]. Considering that older age is a notable risk factor for SAEs [9, 16, 17] owing to several comorbidities [18] or polypharmacy [19], healthcare providers who manage TB must exercise caution while selecting suitable anti-TB drugs and doses and monitor compliance. According to previous studies, pyrazinamide causes the most SAEs among first-line anti-TB drugs [9, 10, 20]. However, there is limited information regarding the safety of PZA in elderly patients [21, 22]. In particular, the incidence and types of SAEs and their relationship with patient age have not been thoroughly investigated previously. Therefore, we aimed to demonstrate the relationship of age and type of SAE to PZA.

## Material and methods

### Study design and patients

We conducted a retrospective cohort study at Seoul National University Bundang Hospital on patients diagnosed with active pulmonary or extra-pulmonary TB who were treated with standard first-line anti-TB medication (INH, RIF, EMB, and PZA) from May 2003 to June 2016. To investigate the time of SAE onset, we excluded patients with treatment failure or death and whose clinical data were insufficient. Additional exclusion criteria were as follows: (i) multi-drug-resistant TB; (ii) non-tuberculosis mycobacterium lung disease; (iii) aged <18 years; (iv) insufficient clinical data owing to transfer, permanent discontinuation of all medications irrespective of cause, loss to follow up, or death; or (v) positive human immunodeficiency virus serology. The analysis of patients who died or discontinued anti-TB therapy is provided in Online Supplementary S1 Table).

This study was conducted in accordance with the amended Declaration of Helsinki [23]. The Institutional Review Board of Seoul National University Bundang Hospital approved the study protocol and waived the need for informed consent owing to the retrospective study design (IRB No. B- 1608-357-113).

### Definition of terms

An SAE was defined as any drug reaction that led to the discontinuation of medication owing to side effects that were intractable to conventional treatment. These events were classifiable as

hepatotoxicity, cutaneous reaction, gastrointestinal (GI) intolerance, or arthralgia [12]. Hepatotoxicity was defined as a transaminase level more than three times greater than normal with associated symptoms such as nausea, vomiting, or abdominal pain or a transaminase level more than five times greater than the upper limit without symptoms. Cutaneous reactions included rash, itching sensation, and eruption, and GI intolerance included nausea, vomiting, and abdominal pain unrelated to hepatotoxicity. Arthropathy included acute gout and articular pain and was clinically diagnosed. The causative drug was determined based on the disappearance of the SAEs upon the withdrawal of the most suspected drug without re-challenge or the recurrence of the same SAE with drug re-challenge. In cases of withdrawal, the decision regarding the discontinuation of the suspected drug was made by the attending physicians [9].

According to the American Thoracic Society guidelines for TB treatment, the standard regimen is defined as a four-drug combination based on body weight as follows: 300 mg/day INH, 10 mg/kg/day RIF, 15–20 mg/kg/day EMB, and 25 mg/kg/day PZA [5]. At least 2 months of the intensive phase and 4 months of the continuation phase should be applied based on current practice guidelines [5].

## Statistical analysis

Data were analyzed using SPSS software for Windows version 25.0 (SPSS Inc., Armonk, New York, USA). The baseline characteristics of the study groups are expressed as the mean and standard deviation (SD). Differences between the 'no SAE' and 'SAE' groups were analyzed using independent t-tests for continuous variables and $\chi^2$ or Fisher's exact tests for categorical variables. Differences were considered significant when $P < 0.05$. In logistic regression, variables were selected based on a univariate analysis. The inclusion criterion for a variable was a significance level of $< 0.1$.

## Results

### Baseline characteristics

In total, 3,384 patients prescribed anti-TB medication were screened, and 2,478 patients diagnosed with pulmonary or extra-pulmonary TB met the study criteria (Fig 1). The median age of the patients was 49.6 years, with pulmonary TB predominating (84.8%), and 53.2% were male. Among the patients, 407 (16.4%) experienced an SAE due to a first-line anti-TB drug during the study period. The median duration of time between the recognition of the SAEs and the first follow-up was 9.0 days (interquartile range, 7.0–14.0 days).

The drug most commonly associated with SAEs was PZA (N = 227, 9.2%), followed by EMB (N = 95, 3.8%), RIF (N = 56, 2.3%), and INH (N = 26, 1.0%) (Fig 1). Among the 227 patients who experienced SAEs due to PZA, 188 (82.8%) and 39 (17.2%) cases were identified by withdrawal and re-challenge, respectively. There were significant differences between patients with SAEs and without SAEs due to PZA with regard to age (49.0 ± 18.3 years vs. 55.5 ± 18.5 years, $P < 0.001$) and treatment duration (7.9 ± 3.1 months vs. 9.2 ± 3.4 months, $P < 0.001$, Table 1). Positive hepatitis B or C serology and baseline liver function tests were not different between the two groups. In cases of death or treatment discontinuation, 44.4% (75/169) of patients experienced a SAE, and the rate of SAEs due to PZA (37.3%) was considerably higher than that due to other drugs (Online Supplementary S1 Table).

### Types of serious adverse events

The incidence rate of hepatotoxicity was the highest (4.7%, 116/2478), followed by cutaneous adverse reaction (3.6%, 89/2478) and GI intolerance (2.9%, 71/2478, Fig 2). According to

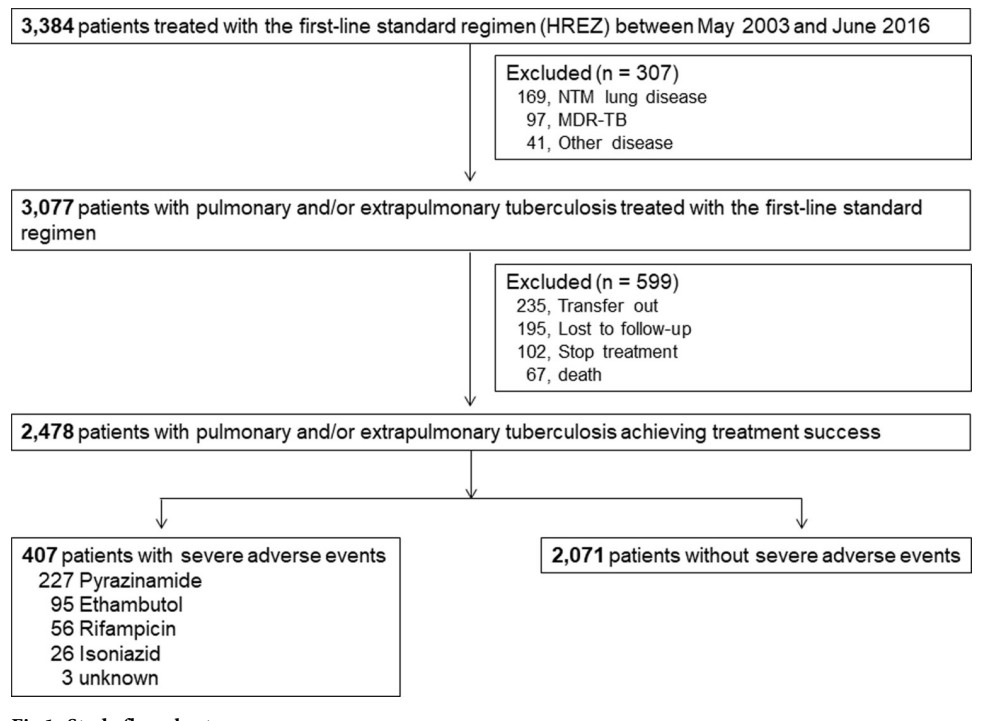

**Fig 1. Study flow chart.**

specific drugs, hepatotoxicity due to PZA was the most common adverse effect, affecting 87.1% of patients (101/116). Cutaneous and GI adverse reactions due to PZA accounted for 40.4% (36/89) and 76.1% (54/71), respectively. Arthropathy, a PZA-specific SAE, occurred in 27 patients. The median onset time of SAEs due to PZA was 22.0 days from treatment initiation, but arthropathy occurred later than the other SAEs (Table 2).

## Risk factors for serious adverse events

Among the patients who experienced SAEs due to the first-line anti-TB drugs, the overall incidence of SAEs was significantly greater with increasing age ($P < 0.001$, Fig 3). These age-incidence relationships were observed with all drugs, except RIF. In the univariate analysis, age and diabetes mellitus (DM) were significant predictive factors for the occurrence of SAEs due to PZA. In the multivariate analysis, age was also identified as an independent risk factor for SAEs due to PZA, when sex and underlying comorbidities, such as chronic liver disease and DM, were adjusted (odds ratio, 1.013; 95% confidence interval [CI], 1.004–1.023; $P = 0.007$, Table 3). Subgroup analysis according to the type of SAE is shown in Online Supplementary S2–S5 Tables. There were significant differences between patients with and without GI intolerance with regard to age, sex, and history of DM, whereas no associated factors were observed in patients with hepatotoxicity, skin rash, and arthropathy.

## Discussion

In this study, adverse effects that led to the interruption or discontinuation of medication occurred in 16.4% of patients, and PZA was the most common causative drug. Although the overall incidence of SAEs increased as age increased, age was not associated with the individual SAEs, except for GI intolerance. Hepatotoxicity was not related to age, which has clinical implications for the treatment of TB. First, although we analyzed only patients who had

**Table 1. Baseline characteristics of the 2484 patients with tuberculosis according to the occurrence of severe adverse events due to pyrazinamide.**

| Variables | Total | Serious adverse events due to PZA | | P value |
|---|---|---|---|---|
| | N = 2478 | (−), N = 2251 | (+), N = 227 | |
| Age (year) | 49.6±18.4 | 49.0±18.3 | 55.5±18.5 | <0.001 |
| Sex, male (%) | 1319 (53.2) | 1048 (46.6) | 111 (48.9) | 0.500 |
| Tuberculosis | | | | 0.291 |
| Pulmonary | 2102 (84.8) | 1904 (84.6) | 198 (87.2) | |
| Extrapulmonary | 376 (15.2) | 347 (15.4) | 29 (12.8) | |
| Initial diagnosis | | | | 0.524 |
| Sputum AFB | 791 (33.7) | 722 (33.9) | 69 (31.8) | |
| TB-PCR | 1553 (66.3) | 1405 (66.1) | 148 (68.2) | |
| Comorbidities | | | | |
| DM | 244 (9.8) | 214 (9.5) | 30 (13.2) | 0.074 |
| Malignancy | 133 (5.4) | 117 (5.2) | 16 (7.0) | 0.238 |
| Renal insufficiency | 26 (1.0) | 20 (1.0) | 4 (1.8) | 0.291 |
| Smoking[a] | | | | 0.718 |
| Never | 1069 (56.8) | 871 (56.6) | 198 (57.7) | |
| Ex- or current | 814 (43.2) | 669 (43.4) | 145 (42.3) | |
| Alcohol use[b] | | | | 0.553 |
| Never/social | 618 (89.7) | 473 (89.2) | 145 (91.2) | |
| Heavy | 71 (10.3) | 57 (10.8) | 14 (8.8) | |
| Long-term steroid use | 4 (0.2) | 4 (0.2) | 0 (0.0) | 1.000 |
| Risk factors for hepatotoxicity | | | | |
| HBs Ag (+) | 57 (4.2) | 51 (4.2) | 6 (4.0) | 0.912 |
| Anti HCV (+) | 12 (0.9) | 12 (1.0) | 0 (0.0) | 0.381 |
| Initial laboratory tests | | | | |
| AST | 22.0±11.8 | 21.8±11.1 | 24.2±16.8 | 0.090 |
| ALT | 19.4±14.6 | 19.4±14.8 | 19.6±13.5 | 0.881 |
| Treatment duration (mo.) | 8.0±3.1 | 7.9±3.1 | 9.2±3.4 | <0.001 |

AFB, acid-fast blue; TB, tuberculosis; PCR, polymerase chain reaction; DM, diabetes mellitus;

[a]Data were not recorded for 595 (24.0%) patients.

[b]Data were not recorded for 1789 (72.2%) patients.

Data are reported as the mean ± standard deviation and numbers (%).

follow-up data from initiation to end of treatment, the incidence of SAEs was not uncommon and would have likely been higher if patients with treatment failure or those who were lost to follow-up had been included. This emphasizes the importance of recognizing SAEs early and highlights the need to modify drug regimens to prevent drug-related morbidity and mortality and improve treatment adherence. Second, because elderly patients are more likely to be vulnerable to anti-TB medication, physicians should be particularly aware of SAEs when treating older patients with TB. Third, the high confirmation rate as a causative drug by only de-challenging might suggest that physicians preferentially discontinued PZA because it was not a maintenance drug and as previously reported side effects were high. Thus, although PZA is an elementary drug and contributes to reducing treatment duration, it is essential to consider its risks and benefits for individual patients when choosing TB medication.

In previous studies, SAEs were most commonly attributable to PZA [9, 10, 20]. Among the SAEs, hepatotoxicity was substantially more common than other types of adverse events. Old age, female sex, treatment regimen, abnormal liver function test at baseline, and previous

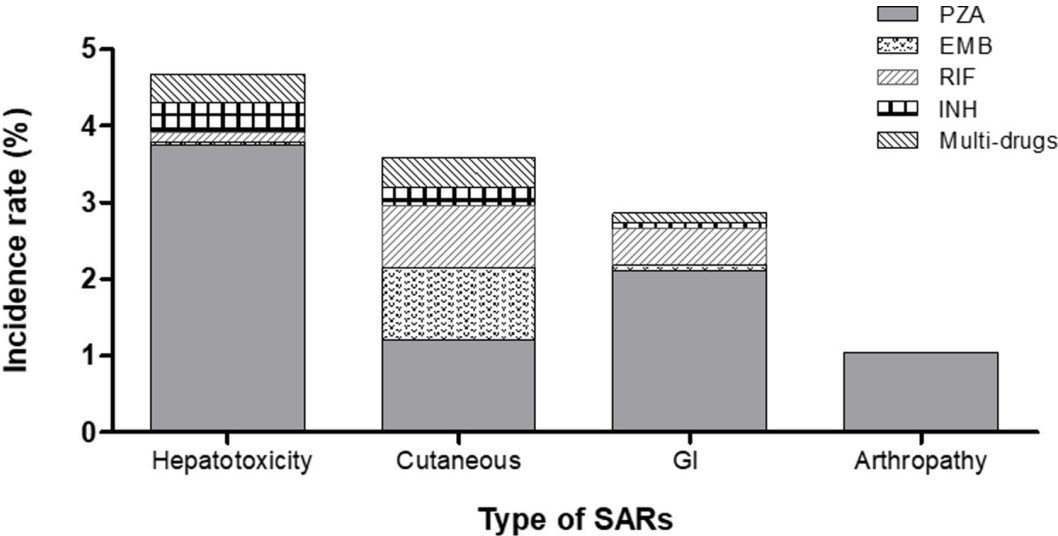

**Fig 2. The proportion of each anti-tuberculosis drug related to severe adverse events.**

history of hepatitis were reported to be risk factors for PZA-associated hepatotoxicity [7, 24, 25]. Although the exact mechanism of hepatotoxicity is not fully understood, drug metabolism rather than the peak drug level might contribute to the development of hepatotoxicity to PZA. For instance, an animal study showed that two PZA metabolites, pyrazinoic acid and 5-hydro-xypyrazinoic acid, are responsible for hepatotoxicity [26]. In contrast, several studies measuring the serum PZA level revealed that the occurrence of hepatotoxicity was not significantly related to the drug concentration [20, 27]. As drug-drug interaction owing to polypharmacy [28] and physiological changes such as decreased hepatic blood flow in the elderly [29] might affect the pharmacodynamics and pharmacokinetics of drugs, elderly patients may be more likely to experience hepatotoxicity. In our study, a relationship between age and the development of hepatotoxicity was not identified. Additionally, a recently published prospective study with 89 patients aged 80 or older showed that a PZA-containing regimen does not increase the rate of hepatotoxicity compared to INH-, RIF-, and EMB-containing regimens, which is consistent with our results [22]. Considering that PZA is a cornerstone drug for the treatment of TB, further studies on biomarkers to predict the occurrence of hepatotoxicity are warranted.

**Table 2. Types and onset time of severe adverse events due to PZA.**

| Variables | No. patients (%) | Time of onset (days, IQR) |
|---|---|---|
| Total | 227 (100.0) | 22.0 (14.0–42.0) |
| Hepatotoxicity | 101 (44.5) | 27.0 (14.0–47.5) |
| GI intolerance | 54 (23.8) | 18.0 (11.8–33.8) |
| Cutaneous adverse reactions | 36 (15.9) | 19.5 (12.0–37.3) |
| Arthropathy | 27 (11.9) | 42.0 (21.0–59.0) |
| Others[†] | 9 (4.0) | 15.0 (12.5–18.5) |

PZA, pyrazinamide; IQR, interquartile range; GI, gastrointestine.

Data are reported as the median (range) and numbers (%).

[†] Paraesthesia in 4 (1.4%), general weakness in 2 (0.7%), central nervous symptoms in 2 (0.7%) and thrombocytopenia in 1 (0.4%).

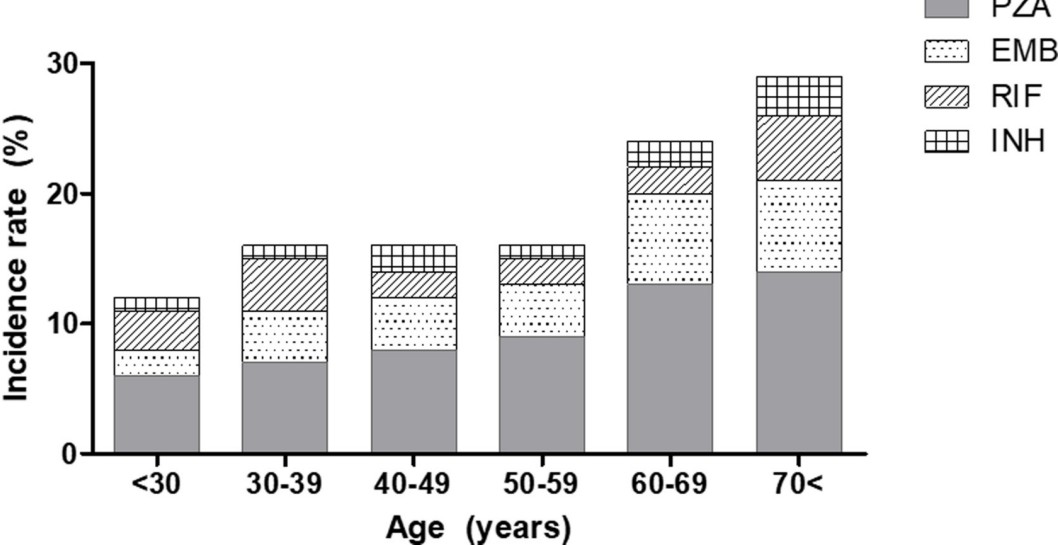

**Fig 3. Incidence of serious adverse events due to pyrazinamide according to age group.**

GI intolerance was the second most frequent SAE due to PZA in our study. Unlike other SAEs, older age was the greatest risk factor for GI intolerance, along with female sex and DM. Kwon et al. reported that GI intolerance occurs at a high rate in the elderly (≥65 years), affecting 10% of the older group [30]. Yee et al. showed that age over 60 is an independent risk factor for GI problems but that this is not the case in those under 35 (hazard ratio, 6.4; 95% CI, 1.2–3.6), which support our results. In contrast, Xiaozhen et al. reported that severe GI disorders occur in less than 1% of the total patients [31]. This study included younger patients, more males, and a lower proportion of patients with DM than our study. These conflicting results might be attributed to demographic factors and underlying medical conditions. As gastroparesis affects 40% of longstanding patients with DM [32], GI disorders due to PZA might be synergistic with DM gastropathy. Therefore, in cases of elderly patients with DM who suffer from GI motility disorders, PZA should be administered with caution.

Cutaneous adverse reactions also represented a frequently observed SAE in our study. Overall, the incidence of cutaneous adverse reactions due to anti-TB medication varies across studies, and the results for the most common causative drugs also differ [9, 33, 34]. For instance, Marra et al. reported that an EMB-containing regimen tends to increase the risk of cutaneous adverse reactions, though there is no statistically significant difference [33]. Similarly, in a study by Kim et al., 16.8% of patients experienced severe cutaneous adverse

**Table 3. Risk factors for the occurrence of severe adverse events due to pyrazinamide.**

| Variables | Univariate | | | Multivariate | | |
|---|---|---|---|---|---|---|
| | OR | 95% CI | P value | OR | 95% CI | P value |
| Age (year) | 1.019 | 1.012–1.027 | <0.001 | 1.013 | 1.004–1.023 | 0.007 |
| Sex, Male (%) | 0.910 | 0.693–1.196 | 0.500 | 1.245 | 0.881–1.760 | 0.214 |
| Chronic liver disease | 0.868 | 0.367–2.051 | 0.747 | 0.944 | 0.398–2.240 | 0.896 |
| DM | 1.450 | 0.963–2.182 | 0.075 | 1.039 | 0.628–1.719 | 0.883 |
| Renal insufficiency | 1.817 | 0.621–5.321 | 0.276 | | | |

OR, odds ratio; CI, confidence interval; DM, diabetes mellitus.

reactions, and SAEs due to EMB were more common (6.2%) than those to PZA (3.7%) [34]. In contrast, Yee et al. reported that the incidence of severe rash is mostly associated with PZA [9]. These inconsistent results might be attributed to differences in treatment duration or other concomitant medications. Although the incidence of PZA-associated cutaneous reactions varies and is more rarely reported than other types of SAEs, physicians should note that PZA also causes cutaneous reactions.

Our study had several limitations. First, the study was conducted at a single center with a retrospective design. Additionally, there were no set protocols for following or managing SAEs. Nonetheless, a small group of physicians were involved in diagnosing and treating TB via a consistent approach to manage side effects; thus, bias was reduced. In addition, analysis based on a large number of subjects confers robust evidence. Second, because we only included patients who had sufficient data to evaluate the SAEs of individual drugs, SAEs in patients who died or discontinued treatment may have been underestimated in terms of frequency. Nevertheless, when death or early discontinuation cases were included, PZA was shown to be the most common SAE-causing drug. Third, data regarding risk factors for hepatotoxicity, such as viral hepatitis or history of alcohol consumption, were missing for a large number of patients. Testing for viral hepatitis is not routinely carried out before anti-TB treatment, and only 55.2% of patients had available data in this analysis. Fourth, although we defined SAEs as cases requiring discontinuation or interruption of anti-TB medication, we did not further classify each SAE by severity. Therefore, further investigation into the association between SAE severity and drug treatment is warranted.

## Conclusions

The incidence of SAEs due to PZA was 9.2%, and old age was a notable risk factor for SAE occurrence. In the majority of cases, a causal relationship was identified by the withdrawal of PZA, even without re-challenge. Therefore, the importance of the early recognition of SAEs and decisions regarding whether to discontinue should be emphasized to ensure effective treatment compliance, especially in elderly patients.

## Supporting information

**S1 Table. Serious adverse events of first-line anti-TB medication in patients who died or discontinued treatment.**
(DOCX)

**S2 Table. Baseline characteristics of patients with pyrazinamide-associated hepatotoxicity.**
(DOCX)

**S3 Table. Baseline characteristics of patients with pyrazinamide-associated GI intolerance.**
(DOCX)

**S4 Table. Baseline characteristics of patients with pyrazinamide-associated skin reaction.**
(DOCX)

**S5 Table. Baseline characteristics of patients with pyrazinamide-associated arthropathy.**
(DOCX)

## Author Contributions

**Conceptualization:** Byoung Soo Kwon, Youlim Kim, Jae Ho Lee.

**Data curation:** Youlim Kim, Sang Hoon Lee, Sung Yoon Lim.

**Formal analysis:** Byoung Soo Kwon, Young-Jae Cho, Jae Ho Lee.

**Investigation:** Yeon Joo Lee, Ho Il Yoon.

**Methodology:** Byoung Soo Kwon, Choon-Taek Lee.

**Resources:** Jong Sun Park.

**Supervision:** Jae Ho Lee.

**Writing – original draft:** Byoung Soo Kwon, Jae Ho Lee.

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
