## [Decision Letter · Decision Letter 0]

29 Apr 2020

PONE-D-20-09108

The high incidence of severe adverse reactions to pyrazinamide in elderly patients with tuberculosis

PLOS ONE

Dear Dr. Lee,

Thank you for submitting your manuscript to PLOS ONE. After careful consideration, we feel that it has merit but does not fully meet PLOS ONE’s publication criteria as it currently stands. Therefore, we invite you to submit a revised version of the manuscript that addresses the points raised during the review process.

We would appreciate receiving your revised manuscript by Jun 13 2020 11:59PM. To enhance the reproducibility of your results, we recommend that if applicable you deposit your laboratory protocols in protocols.io, where a protocol can be assigned its own identifier (DOI) such that it can be cited independently in the future. For instructions see: http://journals.plos.org/plosone/s/submission-guidelines#loc-laboratory-protocols

We look forward to receiving your revised manuscript.

Kind regards,

HASNAIN SEYED EHTESHAM

Academic Editor

PLOS ONE

Journal Requirements:

Additional Editor Comments (if provided):

Major Revision

Reviewers' comments:

Reviewer's Responses to Questions

**Comments to the Author**

1. Is the manuscript technically sound, and do the data support the conclusions?

Reviewer #1: Partly

Reviewer #2: Yes

2. Has the statistical analysis been performed appropriately and rigorously? 

Reviewer #1: I Don't Know

Reviewer #2: Yes

3. Have the authors made all data underlying the findings in their manuscript fully available?

Reviewer #1: Yes

Reviewer #2: No

4. Is the manuscript presented in an intelligible fashion and written in standard English?

Reviewer #1: Yes

Reviewer #2: Yes

5. Review Comments to the Author

Reviewer #1: 1. The methodology of how causality to pyrazinamide was assessed must be described in detail.

2. Number of serious adverse drug reactions identified by drug withdrawal (dechallenge) and the by drug re-introduction (rechallenge) must be presented and discussed.

3. The acronym SARS and SARs have been used for serious adverse drug reactions. In view of the world-wide pandemic related to the SARS-Cov-2 virus infection, the acronym used by the authors has every chance to be misinterpreted. It is suggested that the authors modify this acronym

4. The 'Introduction' is not well developed. Why the present study was necessary to be conducted with the available published literature is not explained.

5. The Discussion section contains repetition of the results. The discussion needs to be more pointed towards the issues identified from the results of the study. A loosely written discussion reads like a review of literature.

6. Please highlight the new knowledge presented in the paper .

Reviewer #2: Comments:

The study by Kwon et al. is conducted very well in retrospectively especially on both pulmonary and extra-pulmonary TB patients; data is well organized and presented. The manuscript is well written and understandable with SARs occurred with PZA, most commonly hepatotoxicity and GI intolerance in elderly patients. Following are some suggestions which can make the manuscript stronger and beneficial for the readers:

Abstract:

• Abbreviation of TB was maintained for “first-line anti-tuberculosis”, correct it.

• Specify the name of first-line anti-tuberculosis drugs.

• Conclusive statement should be modified to give good interpretation to the reader.

Method: It observed that author/s ware collected the data very precisely. However, required some information to improve manuscript like:

• Author should specify the follow-up criteria for EPTB patients during SARs.

Results:

• In patient history, alcoholism and smoker status of patient as well as house hold contact should be noted if possible. Because these habits are strongly associated with TB disease and attributed to high risk of SARs in elderly patients.

• In case of EPTB, SARs analysis may be different according to the TB infection site with the involvement of vital organs other then lung. Author should specify if any association were found in present study or not?

• Author should also provide data of drug resistance patterns other then MDR-TB (already excluded for the study), which may show some other aspects to treat the patients with older regimen with PZA or newer regimen without PZA.

6. PLOS authors have the option to publish the peer review history of their article (what does this mean?). If published, this will include your full peer review and any attached files.

Reviewer #1: Yes: Prof. Nilanjan Saha

Reviewer #2: No

---

## [Author Response · Author response to Decision Letter 0]

3 Jun 2020

We would like to thank you and the reviewers for your comments regarding our manuscript. After incorporating the Reviewer's comments into our manuscript, we believe the new version to be significantly improved. Please see the details in the attached file. We hope that with these changes the manuscript is now suitable for publication in PLOS ONE.

---

## [Editor Report · Decision Letter 1]

30 Jun 2020

The high incidence of severe adverse events due to pyrazinamide in elderly patients with tuberculosis

PONE-D-20-09108R1

Dear Dr. Lee,

We’re pleased to inform you that your manuscript has been judged scientifically suitable for publication and will be formally accepted for publication once it meets all outstanding technical requirements.

Kind regards,

Hasnain Seyed Ehtesham

Academic Editor

PLOS ONE

Additional Editor Comments (optional):

I have gone through the revised manuscript and also the Authors response to reviewers comments. All comments of the reviewers have been satisfactorily addressed by the Authors. Authors have modified the title of the manuscript to reduce the chances of misinterpretation as mentioned by one of the reviewers. Appropriate changes have been made in the Abstract and conclusion section. Revised versions of Table 1 and Table S2-S5 have been included. I recommend this manuscript for publication.
---

## [Editor Report · Acceptance letter]

8 Jul 2020

PONE-D-20-09108R1 

The high incidence of severe adverse events due to pyrazinamide in elderly patients with tuberculosis 

Dear Dr. Lee:

I'm pleased to inform you that your manuscript has been deemed suitable for publication in PLOS ONE. Congratulations! Your manuscript is now with our production department. 

Kind regards, 

on behalf of

Prof Hasnain Seyed Ehtesham 

Academic Editor

PLOS ONE